# Fuzzy Random Chance-Constrained Programming Model for the Vehicle Routing Problem of Hazardous Materials Transportation

**Liying Zhao** [1] and **Ningbo Cao** [2,*]

1   School of Economics and Management, Xi'an University of Technology, Xi'an 710000, China; lyzhao@xaut.edu.cn
2   College of Transportation Engineering, Chang'an University, Xi'an 710000, China;
*   Correspondence: caonb@chd.edu.cn

**Abstract:** As an indispensable necessity in daily routine of citizens, hazardous materials (Hazmat) not only plays an increasingly important role, but also brings a series of transportation uncertainty phenomena, the most prominent of which is a safety problem. When it attempts to find the best vehicle route scheme that can possess the lowest risk attribute in a fuzzy random environment for a single warehouse, the influence of cost should also be taken into account. In this study, a new mathematical theory was conducted in the modeling process. To take a full consideration of uncertainty, vehicle travel distance and population density along the road segment were assumed to be fuzzy variables. Meanwhile, accident probability and vehicle speed were set to be stochastic. Furthermore, based on the assumptions, authors established three chance constrained programming models according to the uncertain theory. Model I was used to seek the achievement of minimum risk of the vehicle route scheme, using traditional risk model; the goal of Model II was to obtain the lowest total cost, including the green cost, and the main purpose of Model III was to establish a balance between cost and risk. To settle the above models, a hybrid intelligent algorithm was designed, which was a combination of genetic algorithm and fuzzy random simulation algorithm, which simultaneously proved its convergence. At last, two experiments were designed to illustrate the feasibility of the proposed models and algorithms.

**Keywords:** hazardous materials; vehicle route model (VRP); uncertainty theory; chance constrained programming model; hybrid intelligent algorithm

---

## 1. Introduction

With the evolution of industrial society, the demand for the logistics industry, especially hazardous materials, which are different from ordinary goods in physical nature and are considered as moving "hazard source" in the transportation process, is constantly increasing. Huge supply demand causes the inter-regional road transportation to be in short supply status and road flow is close to the maximum capacity for a long time. At the same time, public consciousness responding to danger is gradually strengthened, which forced the world to cope with the challenge that hazmat brings. In this condition, any minor uncertainty factor is likely to give rise to risk increment during transportation, therefore, bringing decision-making changes of vehicle routing arrangement. Especially in a situation where uncertain factors change dramatically, random factors and fuzzy influence can easily endanger safety of humans, the environment and ecology, thus, leading to an ascending tendency of risk and cost.

No matter the existing cost-oriented or risk-oriented traditional vehicle, the routing model cannot fully play its role. In a deterministic environment, in accordance with practice, each factor is

treated as a constant variable, the pre-arranged route is unable to deal with the emergency during transportation, in this way it might cause unpredictable consequences. Hence, it is necessary to add uncertain factors in the modeling stage to improve the vehicle routing scheme. Due to the maturity of uncertain theory, deformation period for traditional vehicle routing model must exist for a long time in the process of uncertain theory popularization. Uncertain programming applied to hazmat transportation can be divided into three aspects—random, fuzzy, and fuzzy random programming. If stochastic, it usually refers to accident probability, which follows a particular random distribution, and the solution is focused on how to avoid the occurrence of danger. Whereas, a fuzzy situation uses a fuzzy distribution variable to describe accident consequence concentrating on narrowing the range of influence. Accompanied by the decision makers' risk-averse attitude change and continuous application of uncertainty theory, hazmat transportation accident is considered to be a fuzzy random event, therefore, it is urgent to use new methods to solve fuzzy random programming.

In view of the above requirements, this study will study fuzzy and random factors that occurred in the hazmat transportation, then consider multiple demands of supply chain participants, such as the minimum risk value, which is the ideal state the government hopes to achieve, and the minimum cost, which is the goal enterprise pursued. Hence, exploring and establishing different vehicle routing models through comprehensive cost and risk is of practical value.

The remainder of this study is organized as follows. Related literature on hazardous materials transportation is introduced in Section 2. Section 3 gives a glimpse of preliminaries on uncertain theory. Section 4 describes three models for the vehicle routing problem provided in this study. Section 5 designs an algorithm and its component sub-algorithms. Section 6 discusses the computational experiments and the results. Section 7 contains our final conclusions from the research and provides a set of further research directions.

## 2. Literature Review

As stated by Zografos [1], most research work focused on modifying versions of optimization objectives [2–4], no matter the minimum risk or cost. In this study, the authors are bent on formulating fuzzy random chance-constrained vehicle routing problems (FRCVRP) for hazmat, with the comprehensive consideration of fuzzy random risk and cost. This section reviews related papers on risk assessment methods and uncertain applications for hazmat, respectively.

### 2.1. Literature on Risk Assessment

Due to a lack of standard risk value assessment benchmark, some methods were tested by existing various traditional VRP instances [5]. Erkut and Ingolfsson provided a classification for models of risk calculation, laying a foundation for the study of hazmat [6–8], some of the high frequency used models are accident probability (AP) model, population exposure (PE) model, traditional risk (TR) model, and so on. For example, AP model was adopted by Jia, and it used the probability of a worst case accident to define different road categories with the same accident rate [9]. Li and Leung found that different population values led to different optimal paths in PE model. However, the data on basic resident population is difficult to be accurately acquired [10]. Wei innovatively proposed indeterminate TR model to assess risks at different confidence levels [11]. Additionally, an environmental risk (EN) model is put forward according to the actual scenario, and Cordeiro pointed out that potential risk strongly depended on the nature of the hazmat and presented an approach for assessing environmental risk [12].

### 2.2. Literature on Hazardous Materials Transportation Related to Uncertain Theory

Compared to the classical VRP problem, the research on FRCVRP proposed by Dantzig and Ramser started relatively late (1959) [13]. With uncertain factors becoming the focus of research, simultaneously, the green factor is also getting some attention from researchers. Emrah Demir and Laporte provided a research direction on green road transportation and established a Pollution-Routing

Problem based on VRP [14]. By combining the above two aspects, existing studies on hazardous materials can also be split into three categories—random, fuzzy, and fuzzy random programming.

For the first category, it is assumed that parameters affecting risk or cost are governed by random factors. For instance, Lam explored risk formation mechanism in the liquefied petroleum gas field in Japan, the greatest extent to satisfy consumer acceptable level for incidents, by using a probabilistic network modeling approach [15]. Jabir formulated an integer linear programming model for a capacitated multi-depot green VRP, by integrating economic and emission cost reduction [16]. Bula studied a multi-objective vehicle routing problem and adopted two improved solution methods on the basis of neighborhood search to solve it [17]. The accident probabilities were evaluated according to operators and relevant agencies by Poku-Boansi, from qualitative and quantitative insights, using an instance of Accra–Kumasi Highway (N6) in Ghana [18]. Ghaderi formulated a two-stage stochastic programming model in a multimodal network including transfer spots, with the intention of minimizing transportation cost and risk, considering the location and routing problem [19]. Although it did not distinguish between fuzziness and randomness, Qu pointed out that the risks were relevant to time and route condition, and developed a novel MILP model to build the optimal shipping route with minimal risk [20].

Except for the stochastic parameters used in the modeling process, other research work adopted fuzzy theory, which can describe the transportation scenario more realistically. For example, Ghaleh proposed a pattern of assessing safety risk, by using the Analytical Hierarchy Process, under the fuzzy road fleet transportation scene [21]. Deng addressed fuzzy length between nodes to settle the shortest path problem by using the Dijkstra algorithm [22]. Zero applied triangle fuzzy number to specify cost objective, then expanded this theory to risk objective, and balanced the trade-off between them [23]. Li considered VRP as a nonlinear mono-objective programming rather than a multi-objective programming problem, after dealing with the uncertainty of environment benefits [24]. However, a bi-objective nonlinear integer programming model was established by using triangular fuzzy numbers, to facilitate population exposure from a fuzzy programming prospect by Moon [25]. Hu established a credibility goal programming model aiming at achieving minimum positive deviations value of expected risk and cost from the predefined risk level and cost level, simultaneously [26]. Similarly, in response to multiple depots to customers, Du developed a fuzzy bi-level programming model for the purpose of minimizing the total expected risk and cost under the scenario [27]. He also presented a fuzzy multi-objective programming model that optimizes transportation risk, travel time, and fuel consumption, based on the shortest path mode [28].Triggered by the affected people could be described to be a fuzzy variable, Wei established a chance-constrained programming model, obtaining a balance between risk and cost in the premise of transportation cost, which was also a fuzzy variable [11].

Despite numerous studies related to hazardous materials focusing on stochastic models and fuzzy models separately in the past decade, there are also some studies combining two aspects and proposing a fuzzy random programming model for optimal solutions, with least cost or risk, no matter the route choice problem, the vehicle routing problem, or the location-routing problem. Ma concentrated on how to make uncertain decisions under a different environment of route selection problem, such as fuzzy or stochastic environment, and demonstrated dissimilitude between uncertain and certain scenarios for hazardous materials [29]. Wei firstly assumed that transportation risks were time-dependent fuzzy random variables, and then developed a scheduling optimization model to optimize departure and dwell times for each depot-customer pair [30].

A detailed list and a classification for the hazmat routing problems on uncertain programming are shown in Table 1.

**Table 1.** A summary of routing problem for hazmat.

| No | Author | Stochastic | Fuzzy | Fuzzy Stochastic | Green | R | LR | VRP |
|----|--------|------------|-------|------------------|-------|---|----|-----|
| 1 | Jabir [16] | √ | | | √ | | | √ |
| 2 | Ghaderia [19] | | | | | | √ | |
| 3 | Qu [20] | | | | | √ | | |
| 4 | Deng [22] | | | | | √ | | |
| 5 | Du [28] | | √ | | | √ | | |
| 6 | Wei [11] | | √ | | | | √ | |
| 7 | Wei [30] | | | √ | | | √ | |
| 8 | Ghaffari [31] | | √ | | | | √ | |
| 9 | Xu [32] | | | √ | | | | √ |
| 10 | Hassan-Pour [33] | √ | | | | | √ | |
| 11 | Ji [34] | | √ | | | √ | | |
| 12 | Hu [35] | | √ | | | | √ | |
| 13 | Wang [36] | | √ | | | | | √ |
| 14 | Contreras [37] | √ | | | | | √ | |
| 15 | Mohammadi [38] | √ | | | | | √ | |
| 16 | Samanlioglu [39] | √ | | | | | √ | |
| 17 | Tas [40] | √ | | | | | | √ |
| 18 | Bertazzi [41] | √ | | | | | | √ |
| 19 | Zheng [42] | | √ | | | | | √ |
| 20 | Meng [43] | | √ | | | | | √ |
| 21 | Zhou [44] | | √ | | | | | √ |
| 22 | Present work | | | √ | √ | | | √ |

### 2.3. Research Gap

From the Table 1 presented above, it is apparent that models on uncertain theory for hazardous materials have significant research work on VRP, in the literature. Furthermore, there are a few studies on green elements in fuzzy random programming, because the green VRP problem tends to drop gas emissions to the bottom, and the human value is so difficult to measure that it is usually ignored. However, as an integral part of the total cost, it has a certain importance. The length and speed of the driving route are not a constant value and can change easily within a certain range. They are decided by the driver, so it is reasonable to take human role into account.

On the other hand, fuzzy stochastic programming contains double uncertain attributes, which are probability measure and credibility measure, but most researchers would like to do research just from an angle, mainly because of their different emphasis. Stochastic programming tries to reduce the probability of accidents. Fuzzy measure, as a supplement to random measure, prefers to describe the impact of accidents in language. Language processing is difficult to use in mathematical theory to do accurate calculation, so there is less research in this area.

Hence, it is essential to solve the VRP problem by considering risk and total cost, including the green factor in a realistic scenario. The proposed model deals with modeling and analysis for vehicle routing problem under an uncertain environment. To the best of our knowledge, in this regard, this paper is pioneer study on multi-modal VRP, using chance measure, by considering the risks and the total costs including green costs. Two instances of model-orientation were figured out by hybrid intelligence algorithm, which combined genetic algorithm and fuzzy random simulation algorithm. Thus, the establishment and analysis of three models for the vehicle routing problem are the main contributions of the research presented in this paper.

## 3. Preliminaries

Although both fuzziness and randomness belong to uncertainty, they can easily be confused. They are two distinct concepts. The fuzzy event can be described by credibility measure via a membership function, after being pioneered by Zadeh [45]. However, random events usually use probability measure to calculate the probability of occurrence. Fuzziness works as a complementary

role for randomness, there are similarities between the two mathematical terms. The membership function in fuzzy theory is analogous to probability density function in random theory, similarly, credibility measure defined by Li and Liu is parallel to probability measure from a theoretical point of view [46]. As risk and cost happening during hazmat transportation have dual attribute, a VRP model using chance measure was considered in this study; a detailed introduction on fuzzy random theory is first presented.

*3.1. Fuzzy Theory*

**Definition 1** ([47]). *Let $\Theta$ be a nonempty set, and $P(\Theta)$ is the power set of $\Theta$, for each $A \in P(\Theta)$, there is a nonnegative number $Pos(A)$, called its possibility, such that*

*(i)*    *$Pos\{\emptyset\} = 0$, $Pos\{\Theta\} = 1$ and,*
*(ii)*   *$Pos\{\cup_k A_k\} = \sup_k Pos(A_k)$ for any arbitrary collection $\{A_k\}$ in $P(\Theta)$*

The triplet $(\Theta, P(\Theta), Pos)$ is called a possibility space, and the function $Pos$ is referred to as a possibility measure.

**Definition 2** ([48]). *Let $\xi$ be a fuzzy variable on a possibility space $(\Theta, P(\Theta), Pos)$. Then its membership function is derived from the possibility measure Pos by*

$$\mu(x) = Pos\{\theta \in \Theta | \xi(\theta) = x\}, x \in R$$

**Definition 3** ([49]). *Let $(\Theta, P(\Theta), Pos)$ be a possibility space, and A be a set in $P(\Theta)$. Then the necessity measure of A is defined by*

$$Nec\{A\} = 1 - Pos(A^c)$$

**Definition 4** ([50]). *Let $(\Theta, P(\Theta), Pos)$ be a possibility space, and A be a set in $P(\Theta)$. Then the credibility measure of A is defined by*

$$Cr\{A\} = (pos\{A\} + Nec\{A\})/2$$

If the membership function $\mu()$ of is $\xi$ given as $\mu$, then the possibility, necessity, credibility of the fuzzy event $\{\xi \geq r\}$ can be represented, respectively, by

$$Pos\{\xi \geq r\} = \sup_{\mu \geq r} \mu(\mu), Nec\{\xi \geq r\} = 1 - \sup_{\mu < r} \mu(\mu)$$

$$Cr\{\xi \geq r\} = \{Pos(\xi \geq r) + Nec(\xi \geq r)\}/2$$

**Definition 5** ([51]). *Let $\xi$ be a fuzzy variable, then the function given below $\Phi: (-\infty, +\infty) \rightarrow [0,1]$, $\Phi(x) = Cr\{\theta \in \Theta | \xi(\theta) \leq x\}$ is called the credibility distribution of fuzzy variables $\xi$.*

**Example 1.** *A trapezoidal fuzzy variable $\xi = (r_1, r_2, r_3, r_4)$ is defined by the following membership function (See Figure 1), then credibility distribution function is given as follows. (See Figure 2).*

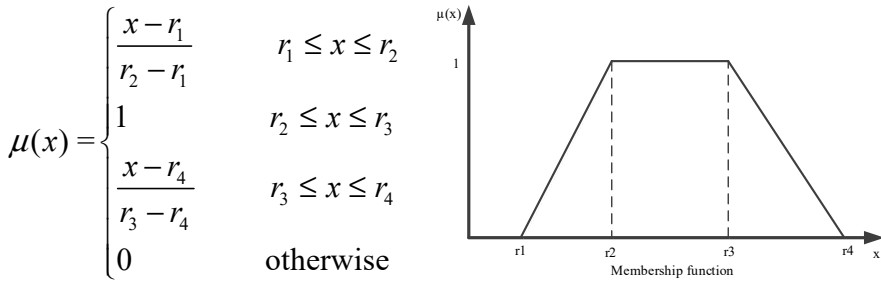

**Figure 1.** Membership for a trapezoidal fuzzy variable.

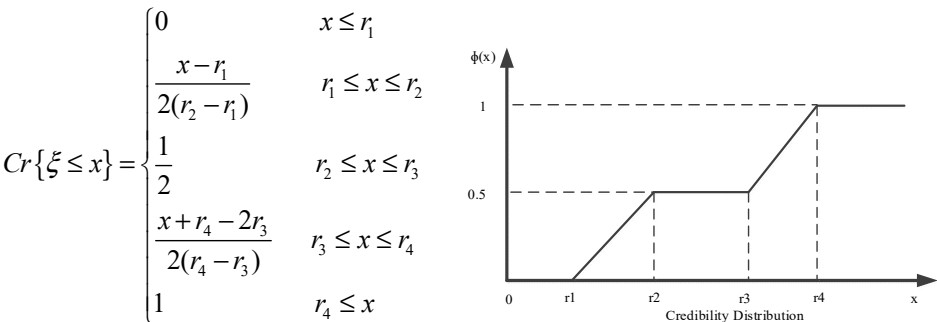

**Figure 2.** Credibility function for a trapezoidal fuzzy variable.

### 3.2. Fuzzy Random Theory

**Definition 6** ([52])**.** *Suppose $\xi$ is a function from probability space $(\Omega, A, Pr)$ to the fuzzy set of variables, if for any Borel set B, $Pos\{\xi(\omega) \in B\}$ is a measurable function about $\omega$, then $\xi$ is called a fuzzy random variable.*

**Example 2.** *Suppose $(\Omega, A, Pr)$ is set as the probability space, if $\Omega = \{\omega_1, \omega_2, \cdots, \omega_m\}$, and $u_1, u_2, \cdots, u_m$ is a fuzzy variable, then*

$$\xi(\omega) = \begin{cases} u_1 & \omega = \omega_1 \\ u_2 & \omega = \omega_1 \\ \vdots \\ u_m & \omega = \omega_1 \end{cases}$$

*is a fuzzy random variable.*

**Example 3.** *Suppose $\eta$ is a random variable in probabilistic space $(\Omega, A, Pr)$, u is a fuzzy variable, and $\xi(\omega) = \eta(\omega)u, \forall \omega \in \Omega$. If for any Borel set B, $Pos\{\xi(\omega) \in B\}$ is a measurable function about $\omega$, then $\xi = \eta u$ is called a fuzzy random variable.*

**Definition 7** ([53,54])**.** *Suppose $\xi$ is a random variable in a probabilistic space $(\Omega, A, Pr)$, B is the Borel set of R, then call the function from $(0, 1]$ to $[0, 1]$*

$$Ch\{\xi \in B\}(\alpha) = \sup_{Pr\{A\} \geq \alpha} \inf_{\omega \in A} Cr\{\xi(\omega) \in B\}$$

*as a chance measure of fuzzy random event $\xi \in B$.*

**Definition 8** ([55])**.** *Suppose $\xi$ is a random variable, and $\gamma, \delta \in (0, 1]$, then*

$$\xi_{inf}(\gamma, \delta) = inf\{r | Ch\{\xi \leq r\}(\gamma) \geq \delta\}$$

*is called $(\gamma, \delta)$-Pessimistic value of $\xi$.*

*3.3. Chance-Constrained-Programming Model*

**Definition 9** ([55]).  *Assume that $x$ is a decision vector, $\xi$ is a fuzzy random vector, $f(x, \xi)$ is the return function, and $g_i(\mathbf{x}, \xi)$ are constraint functions, $i = 1, 2, \ldots, p$. It is obvious that the following*

$$
\begin{cases}
\max \overline{f} \\
\text{subject } to \\
Ch\{f(\mathbf{x}, \xi) \geq \overline{f}\}(\gamma) \geq \delta \\
Ch\{g_i(\mathbf{x}, \xi) \leq 0\}(\alpha_i) \geq \beta_i, i = 1, 2, \ldots, p
\end{cases}
$$

*is a joint chance constraint programming model, where $\alpha, \beta$ and $\gamma, \delta$ are the predetermined confidence levels. Generally, the authors only considered* values $\geq 0.5$.

From the above model, the standard stochastic chance constraint and fuzzy chance constraint programming model could be derived, that is, the chance constraint programming model must contain two measures, a probability measure and a credibility measure, respectively.

## 4. Vehicle Routing Model Formulation

In this section, three mathematical models are proposed whose object goals are different from each other. An explicit description of unified symbols and assumptions is first presented, followed by a discussion related to model formulation and applicability. From the literature, a chance-constrained programming model was conducted that for the vehicle routing problem with green factor, which had three challenges, listed as follows:

(i)     Risk assessment: Due to a series of environmental and human factors, the occurrence of hazardous materials accident is a random event, and the exact consequence of hazardous materials accident was difficult to estimate in advance. Due to the lack of sufficient data, uncertain theory had to be used to solve this intractable problem.

(ii)    Cost calculation: This involved determining different components of total cost, especially green cost, and the biggest difference in this study was considering the distance and speed as uncertain factors from a conventional model.

(iii)   Vehicle routing assignment: This required arranging sequence serving a set of customers assigned to a vehicle under uncertain environment.

Thus, the comprehensive chance-constrained vehicle routing problem encompassing the above-mentioned three aspects aimed to achieve goals, respectively. In this paper, the following three models subjected to uncertain scenario were formulated as a deformation of classical VRP.

Model I: Vehicle routing model for minimum risk—The objective function was to minimize the risk incurred in all sections of routes.

Model II: Vehicle routing model for cost minimum—Analogous to Model I, the objective function of this model was to minimize total cost consumed along routes.

Model III: Integrated model for risk and cost minimization—The objective function of this model was to minimize the equilibrium value between risk and cost from origin to destination node.

The assumptions, notations and decision variables used in the three mathematical formulations are described below.

Assumptions:

i.      The transportation network only has one depot but a set of customers; meanwhile, all vehicles are the same type;

ii.     The number of vehicle fleet is decided by the depot; each vehicle has a physical limitation, i.e., capacity, meaning the sum upload amount of all customers shared a path that cannot exceed it;

iii. A customer must be served and visited once and only once, and transportation time can meet all customer's time window limit;

iv. A vehicle routing scheduling must be a loop circle, beginning from the depot and ending at the same depot;

v. A vehicle can visit an uncertain number of customers only if it is within capacity limitation, if not, the vehicle routing scheduling must be abandoned;

vi. The length of arc and population density along the arc are assumed to be fuzzy variables, this work uses triangle fuzzy variables to describe them;

vii. The hazmat accident probability is in a random format, similarly, the speed of the vehicle is adjustable, which is also a random variable.

viii. The customer demands, including time and amount are known, at least, one day earlier.

The notations for the models are described in Table 2.

**Table 2.** Unified notation for the models.

| **Set:** | |
|---|---|
| $N = (A, V)$ | Transportation network |
| $A = \{0, 1, 2, \ldots, n\}$ | Node set, node 0 denotes single depot |
| $A_0 = A/\{0\}$ | A set of customers waiting for delivery. |
| $V = \{i, j \mid i, j \in A, i \neq j\}$ | A collection of arcs that have been connected between customers. |
| $K = \{1, 2, 3, \ldots k\}$ | A collection of vehicles of the same type available in a depot. |
| **Indices:** | |
| $i, j$ | Customer index |
| $m$ | Depot index |
| $v$ | Vehicle index |
| **Parameter:** | |
| $q_i$ | Demand of customer $i$ |
| $Q$ | Capacity of the vehicle. |
| $w$ | Vehicle weight (empty weight). |
| $F_{fix}$ | Fixed cost for a vehicle. |
| $F_{fuel}$ | Variable vehicle operating cost per unit distance. |
| $F_{emi}$ | $CO_2$ emission cost per unit weight of vehicle per unit |
| $\lambda_{ij}$ | Affected area of the accident on arc $(i, j)$. |
| **Fuzzy parameters:** | |
| $\overline{\xi_{ij}}$ | Length of arc $(i, j)$. |
| $\overline{\rho_{ij}}$ | Average population density along arc $(i, j)$. |
| **Random parameters:** | |
| $p_{ij}$ | Probability of accident occurring on arc $(i, j)$. |
| $v_{ij}$ | Speed of vehicle traveling across arc $(i, j)$. |
| **Decision variables:** | |
| $x_{ij}^k$ | it takes value 1 if arc $(i, j)$ uses vehicle $k$ to travel, it takes value 0, otherwise. |
| $y_i^k$ | it takes value 1 if customer $i$ uses vehicle to travel, it takes value 0, otherwise. |

The three proposed mathematical models adopting the above-mentioned notions are explained from Sections 4.1–4.4.

*4.1. Model I-Vehicle Routing Model for Risk Reduction*

Minimize total risk

$$\overline{R_{sum}} = \sum_{\forall (i,j) \in N} \sum_{\forall k \in K} \overline{R_{ij}} x_{ij}^k \tag{1}$$

where $\overline{R_{sum}}$ denotes the total risk, $\overline{R_{ij}}$ is risk on arc $(i, j)$.

According to Erkut [8], the risk along arc $(i, j)$ can be demonstrated as Figure 3. The affected area is seen as a circle along arc $(i, j)$, with the radius of $r_{ij}$ and a center dot of $k$.

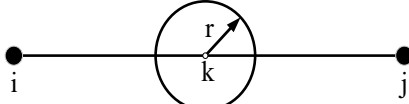

**Figure 3.** Risk along arc $(i, j)$.

Then, the formulation of hazardous materials risk can be expressed as:

$$\overline{R_{ij}} = p_{ij} \cdot \overline{C_{ij}} \tag{2}$$

where $p_{ij}$ means probability of an incident on road segment $(i, j)$ and $\overline{C_{ij}}$ is population consequence along road segment $(i, j)$, thus, $\overline{C_{ij}}$ can be resulted by the product of population density $\overline{\rho_{ij}}$ and affected area $\lambda_{ij}$ of accident happening on arc $(i, j)$. Therefore,

$$\begin{aligned}\overline{R_{sum}} &= \sum_{\forall(i,j)\in N}\sum_{\forall k\in K}\overline{R_{ij}}x_{ij}^k = \sum_{\forall(i,j)\in N}\sum_{\forall k\in K}p_{ij}\cdot\overline{C_{ij}}\,x_{ij}^k \\ &= \sum_{\forall(i,j)\in N}\sum_{\forall k\in K}p_{ij}\cdot\overline{\rho_{ij}}\cdot\lambda_{ij}\,x_{ij}^k = \sum_{\forall(i,j)\in N}\sum_{\forall k\in K}p_{ij}\cdot\overline{\rho_{ij}}\cdot\pi r_{ij}{}^2\cdot x_{ij}^k\end{aligned} \tag{3}$$

From the assumptions and the above equation, the population density is a fuzzy variable. After multiplying area, the number of people affected is still a fuzzy variable. Since accident probability is a stochastic variable, according to Definition 6, the result of risk is a fuzzy random variable.

As we all know, risk $\xi$ is a fuzzy random variable. Suppose $\Omega$ denotes the accident probability set, $(\Omega, A, \text{Pr})$ is probability space, $\overline{C_{ij}} = (200, 250, 280)$ is the accident consequence, then, the risk occurred on arc $(i, j)$, $\xi_{ij}$ can be expressed as follows,

$$\xi_{ij}(\omega) = \begin{cases} (200, 250, 280) & \omega = p_{ij} \\ 0 & \omega = 1 - p_{ij} \end{cases} \tag{4}$$

According to **Definitions 7** and **8**, $\overline{R_{ij}}$ is defined as follows,

$$\begin{aligned}\overline{R_{ij}}\,(\beta, \alpha) &= \left\{ \overline{R} \,\middle|\, Pr\left\{ Cr(p_{ij}\cdot\overline{\rho_{ij}}\cdot\lambda_{ij} \le \overline{R}\,) \ge \beta \right\} \ge \alpha \right\} \\ &= Ch\left\{ p_{ij}\cdot\overline{\rho_{ij}}\cdot\lambda_{ij} \le \overline{R} \right\}(\beta) \ge \alpha\end{aligned} \tag{5}$$

Then the sum risk under the chance measure $(\beta, \alpha)$ is calculated by

$$\begin{aligned}min\,\overline{R_{sum}}(\beta, \alpha) &= min \sum_{\forall(i,j)\in N}\sum_{\forall k\in K}\overline{R_{ij}}\,(\beta, \alpha)x_{ij}^k \\ &= \sum_{\forall(i,j)\in N}\sum_{\forall k\in K}(inf\,Ch\{p_{ij}\cdot\overline{\rho_{ij}}\cdot\lambda_{ij} \le \overline{R}\}(\beta) \ge \alpha)x_{ij}^k \\ &= \sum_{\forall(i,j)\in N}\sum_{\forall k\in K}\overline{R_{ij(inf)}}(\beta, \alpha)x_{ij}^k\end{aligned} \tag{6}$$

### 4.2. Model II-Vehicle Routing Model for Cost Reduction

In this section, authors place emphasis on transportation cost of hazmat with the vehicle routing model, the total cost can be broken up into three components, namely fixed cost, fuel cost, and emission cost [20], which can be expressed as Equation (7).

Minimize total cost

$$\overline{F_{sum}} = F_{fix} + F_{fuel} + F_{emi} \tag{7}$$

The first section means expenses that the propritor must pay during a certain period, which is not related to the transportation business volume. It includes the basic salary and fixed allowance of the worker, enterprise management fee and vehicle depreciation, respectively, that is,

$$F_{fix} = c \cdot \sum_{\forall (i,j) \in N} \sum_{\forall k \in K} x_{ij}^k \tag{8}$$

where parameter $c$ denotes the transformed money of using a vehicle one time servicing a customer.

As for the $F_{fuel}$, in this work, it refers to the following fuel consumption model [28,54],

$$P_t = Mav + Mgvsin\theta + 0.5C_dS\zeta v^3 + MgvC_h cos\theta \tag{9}$$

where $P_t$ represents the total tractive power in watts, $M$ is the total quality of vehicle (curb weight plus carried load). Take gas transportation, for example, explanation and value for parameters used in Equation (9) are listed as follows:

Authors assume that the vehicle travels through a given arc $(i,j)$ at the speed $v_{ij}$, then the total quantity $F_{ij}$ of energy consumed on arc can be approximated as:

$$
\begin{aligned}
F_{ij} &= P_t \times (\overline{\xi_{ij}}/v_{ij}) = (M_{ij}av_{ij} + M_{ij}gv_{ij}sin\theta + 0.5C_dS\zeta v_{ij}^3 + M_{ij}gv_{ij}C_h cos\theta) \times (\overline{\xi_{ij}}/v_{ij}) \\
&= (M_{ij}a + M_{ij}gsin\theta + 0.5C_dS\zeta v_{ij}^2 + M_{ij}gC_h cos\theta)\overline{\xi_{ij}} \\
&= M_{ij}(a + gsin\theta + gC_h cos\theta)\overline{\xi_{ij}} + 0.5C_dS\zeta v_{ij}^2\overline{\xi_{ij}} \\
&= (M_{ij}\phi + \varphi v_{ij}^2)\overline{\xi_{ij}}
\end{aligned}
\tag{10}
$$

where $\phi = a + gsin\theta + gC_h cos\theta$, $\varphi = 0.5C_dS\zeta$, $M_{ij} = w + q_{ij}$, from the above equation, authors can learn that the fuel consumption is related to two factors, not only the mass, but also victory.

Thus, knowing that the value of unit fuel price will provide some convenient method regarding the value of the total fuel cost, it can be calculated as:

$$F_{fuel} = \sum_{\forall (i,j) \in N} \sum_{\forall k \in K} (M_{ij}\phi + \varphi v_{ij}^2)\overline{\xi_{ij}}x_{ij}^k \times \frac{1}{\vartheta} \times P_{fuel} \tag{11}$$

where $\vartheta$ is fuel efficiency and $P_{fuel}$ is fuel price per unit, such as $P_{fuel} = 7\,RMB/L$.

Last but not least, the emission cost is greatly affected by the fuel, $\eta_c$ is fuel conversion factor, $t_c$ is carbon tax, it can define the emission cost as below in Equation (12):

$$F_{emi} = \sum_{\forall (i,j) \in N} \sum_{\forall k \in K} (M_{ij}\phi + \varphi v_{ij}^2)\overline{\xi_{ij}}x_{ij}^k \times \frac{1}{\vartheta} \times \eta_c \times t_c \tag{12}$$

According to the above analysis (7)~(12), the total cost of the vehicles can be aggregated as:

$$
\begin{aligned}
\overline{F_{sum}} &= c \cdot \sum_{\forall (i,j) \in N} \sum_{\forall k \in K} x_{ij}^k + \sum_{\forall (i,j) \in N} \sum_{\forall k \in K} (M_{ij}\phi + \varphi v_{ij}^2)\overline{\xi_{ij}}x_{ij}^k \times \frac{1}{\vartheta} \times P_{fuel} + \sum_{\forall (i,j) \in N} \sum_{\forall k \in K} (M_{ij}\phi + \varphi v_{ij}^2)\overline{\xi_{ij}}x_{ij}^k \times \frac{1}{\vartheta} \times \eta_c \times t_c \\
&= c \cdot \sum_{\forall (i,j) \in N} \sum_{\forall k \in K} x_{ij}^k + \sum_{\forall (i,j) \in N} \sum_{\forall k \in K} (M_{ij}\phi + \varphi v_{ij}^2)\overline{\xi_{ij}}x_{ij}^k \times \frac{1}{\vartheta} \times (P_{fuel} + \eta_c \cdot t_c)
\end{aligned}
\tag{13}
$$

From the assumption, it is known that the length is a fuzzy and the vehicle speed is a random variable. Similar to risk calculations, the sum cost of fuel is also a fuzzy random variable. Then, the sum cost under chance measure $(\chi, \gamma)$ is calculated by,

$$\overline{F_{sum}}(\chi, \gamma) = \left\{ Pr\{Cr(\sum_{\forall (i,j) \in N} \sum_{\forall k \in K} \left[ (M_{ij}\phi + \varphi v_{ij}^2)\overline{\xi_{ij}} x_{ij}^k \times \tfrac{1}{9} \times (P_{fuel} + \eta_c \cdot t_c) \right] \leq \overline{F_{ij}}) \geq \chi \} \geq \gamma \right\} + c \cdot \sum_{\forall (i,j) \in N} \sum_{\forall k \in K} x_{ij}^k$$
$$= Ch\{ \sum_{\forall (i,j) \in N} \sum_{\forall k \in K} \left[ (M_{ij}\phi + \varphi v_{ij}^2)\overline{\xi_{ij}} \times \tfrac{1}{9} \times (P_{fuel} + \eta_c \cdot t_c) \leq \overline{F_{ij}} \right](\chi) \geq \gamma \} x_{ij}^k + c \cdot \sum_{\forall (i,j) \in N} \sum_{\forall k \in K} x_{ij}^k \tag{14}$$

Therefore, the objective of Model II is as follows:

$$min \; \overline{F_{sum}}(\chi, \gamma) = min \; Ch\{ \sum_{\forall (i,j) \in N} \sum_{\forall k \in K} \left[ (M_{ij}\phi + \varphi v_{ij}^2)\overline{\xi_{ij}} \times \tfrac{1}{9} \times (P_{fuel} + \eta_c \cdot t_c) \leq \overline{F_{ij}} \right](\chi) \geq \gamma \} x_{ij}^k + c \cdot \sum_{\forall (i,j) \in N} \sum_{\forall k \in K} x_{ij}^k$$
$$= \sum_{\forall (i,j) \in N} \sum_{\forall k \in K} inf \; Ch\{ \left[ (M_{ij}\phi + \varphi v_{ij}^2)\overline{\xi_{ij}} \times \tfrac{1}{9} \times (P_{fuel} + \eta_c \cdot t_c) \leq \overline{F_{ij}} \right](\chi) \geq \gamma \} x_{ij}^k + c \cdot \sum_{\forall (i,j) \in N} \sum_{\forall k \in K} x_{ij}^k \tag{15}$$
$$= \sum_{\forall (i,j) \in N} \sum_{\forall k \in K} \overline{F_{ij(inf)}}(\chi, \gamma) x_{ij}^k + c \cdot \sum_{\forall (i,j) \in N} \sum_{\forall k \in K} x_{ij}^k$$

### 4.3. Model III-Vehicle Routing Model for Risk and Cost Minimization

In order to work out this integrated model, it can take a compromise value by weighting sum of the two aspects. First, the authors perform the normalization operation on risk and cost. Denote $\overline{R_{(inf)max}}(\beta, \alpha)$, $\overline{R_{(inf)min}}(\beta, \alpha)$ as the maximum and minimum values for total risk under the chance measure $(\beta, \alpha)$, $\overline{F_{(inf)max}}(\chi, \gamma)$, $\overline{F_{(inf)min}}(\chi, \gamma)$, as the maximum and minimum values for total cost, the chance measure $(\chi, \gamma)$, respectively, then the normalized risk and cost are

$$R' = \frac{\overline{R_{(inf)}}(\beta, \alpha) - \overline{R_{(inf)min}}(\beta, \alpha)}{\overline{R_{(inf)max}}(\beta, \alpha) - \overline{R_{(inf)min}}(\beta, \alpha)} \quad F' = \frac{\overline{F_{(inf)}}(\chi, \gamma) - \overline{F_{(inf)min}}(\chi, \gamma)}{\overline{F_{(inf)max}}(\chi, \gamma) - \overline{F_{(inf)min}}(\chi, \gamma)} \tag{16}$$

With the given parameter $\tau \in [0, 1]$, the compromise objective value is

$$T = \tau F' + (1 - \tau)R' \tag{17}$$

### 4.4. Model Constraints

Equation (18) means that customer $j$ is visited by the vehicle $k$, and vehicle $k$ must arrive at the customer $j$ from customer $i$. Equation (19) signifies that customer $i$ can be served by the vehicle $k$, and the vehicle $k$ must arrive at the customer $j$, after delivering the materials from the customer $i$. Equation (20) represents that the load of every vehicle could not exceed the maximum capacity $Q$. Constraint (21) means that each customer must be served by only one vehicle, and constraint (22) means that all vehicle routing arrangements start from the same depot.

$$\sum_{k=1}^{v} x_{ij}^k = y_j^k \;\; \forall k = 1, 2, \ldots v, \;\; \forall j = 1, 2, \ldots, n \tag{18}$$

$$\sum_{k=1}^{v} x_{ij}^k = y_i^k \;\; \forall k = 1, 2, \ldots v, \;\; \forall i = 1, 2, \ldots, n \tag{19}$$

$$\sum_{i=0}^{n} y_i^k \times q_i \leq Q \;\; \forall k = 1, 2, \ldots v \tag{20}$$

$$\sum_{k=1}^{v} y_i^k = \forall i = 1, 2, \ldots n \tag{21}$$

$$\sum_{k=1}^{v} y_0^k = v \tag{22}$$

## 5. Solution Methodology

As stated above, a fuzzy random simulation must be used to acquire the objective value according to the corresponding models. After this, a genetic algorithm on the basis of fuzzy random simulation algorithm is designed to optimize vehicle routing strategy, with the above three models.

### 5.1. Fuzzy Random Simulation Algorithm

Let $\xi$ be an $n$-dimensional fuzzy vector with the membership degree $u_i$ for all $i = 1, 2, \ldots, N$, and let $f : R^n \to R$ be a real function. Then, the credibility $Cr\{\xi \leq r\}$ value can be obtained by

$$L(r) = \frac{1}{2} \left( \max_{f(\mathbf{y}_i) \leq r} u_i + 1 - \max_{f(\mathbf{y}_i) > r} u_i \right)$$

Considering that $L(r)$ is an increasing function, the chance measure value can be calculated by fuzzy random simulation [55], which is the combination of fuzzy simulation and random simulation, after using fuzzy simulation to obtain a series of $\beta$-pessimistic value, then taking the $\alpha$-proportion incremental value to be the approximate cutoff value. The steps of calculating $\beta$-pessimistic value by fuzzy simulation invented by Liu is described as follows [54]:

Step 1.  Initialize a small real number $\varepsilon > 0$.
Step 2.  Randomly generate vectors $\mathbf{y_i}$ with membership degrees $u_i$ for all $i = 1, 2, \ldots, N$.
Step 3.  Calculate the minimum and maximum values $a = min\{f(\mathbf{y_i}) | 1 \leq i \leq N\}$ and $b = max\{f(\mathbf{y_i}) | 1 \leq i \leq N\}$.
Step 4.  Set $r = (a + b)/2$.
Step 5.  If $L(r) \geq \alpha$, set $b = r$. Otherwise, set $a = r$.
Step 6.  If $b - a \geq \varepsilon$, go to Step 4.
Step 7.  Return $(a + b)/2$ as an approximation of the $\beta$-pessimistic value.

Then, the fuzzy random simulation algorithm can be summarized as follows:

Step 1.  Generate $\omega_1, \omega_2, \ldots \omega_N$, from space $\Omega$ according to the probability measure Pr.
Step 2.  Find the smallest values $\overline{f_n}$ such that $Cr\{f(\mathbf{x}, \xi(\omega_n)) \leq \overline{f_n}\} \geq \beta$ for $n = 1, 2, \ldots, N$ by fuzzy simulation, respectively.
Step 3.  Set $NN$ value, which equals to the integer part of $\alpha N$.
Step 4.  Return the $NN$ th largest element in $\{\overline{f}_1, \overline{f}_2, \ldots, \overline{f}_N\}$.

### 5.2. Fuzzy Random Simulation Based Genetic Algorithm

The general procedures of genetic algorithm are initialization, evaluation, selection, crossover, and mutation, in turn.

#### 5.2.1. Initialization Operation

In general, the vehicle routing problem is a combinatorial optimization problem, so the chromosomes can be encoded as integers. Their structures can be divided into two parts, Part C and Part V, so the length is decided by the number of customers and vehicles. Part C stands for the information about the customer order, and Part V denotes vehicle information on how to service several customers using a common vehicle (see Figure 4).

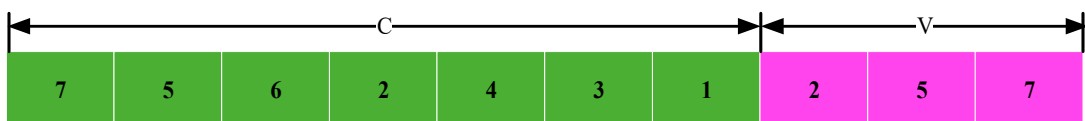

**Figure 4.** Initialization operation.

For example, Part C is initialized to be 7,5,6,2,4,3,1, which means that the service order is 7,5,6,2,4,3,1 in sequence. The content of Part V is the last customer index of Part C in the transportation loop by a common vehicle. For example, there are three vehicles waiting for transportation, so the length of Part V is 3. If Part V list is 2,5,7, it means that the first vehicle serves two customers—customer 7 and customer 5. The second vehicle serves from the next node to the node that is the second number in Part V, node 5, hence, the customer 6, customer 2, and customer 4 use the second vehicle in common. In the same way, the third vehicle serves from the sixth node to the seventh node, which is customer 3 and customer 1. From the above description, it requests the Part V must be in an ascending form.

### 5.2.2. Evaluation Function

Let $Y = (C, V)$ be a chromosome from feasible space. By employing fuzzy random simulation, the objective value can be easily obtained for the three models. From the objective values of three models, it seeks for minimum, so $\frac{1}{objective\ value}$ is defined as the evaluation function Eval (Y).

### 5.2.3. Selection Operation

The aim of selection operation is to select better chromosomes to be parent, and roulette wheel selection method uses fitness-proportions to make choice. This paper adopted this method for the selection operation. First, it calculated the cumulative probability $p_k$ for each chromosome $Yk$, $k = 1, 2, \ldots, popsize$:

$$p_0 = 0, p_k = \sum_{q=1}^{k} Eval(Y_q),\ k = 1, 2, \ldots, popsize$$

Then, compare the randomly generated number $r \in (0, p_{popsize}]$ with the cumulative probability $[p_{k-1}, p_k]$, if $r \in [p_{k-1}, p_k]$, the select chromosomes $Y_k$ to be the parent chromosome.

### 5.2.4. Crossover Operation

Repeat the following scheme for $k = 1, 2, \ldots, popsize$: randomly generated number $r \in (0, 1]$, if $r$ is lower than the predefined crossover probability $p\_c$, the corresponding chromosome is chosen to be a parent.

The crossover operation mainly acts on part V, take the chromosome pair $(Ya, Yb)$, for example, it changes the whole Part V, such as the chromosome $Ya$ are coded as 7,5,6,2,4,3,1,2,5,7, $Yb$ are coded as 2,5,4,3,7,1,6,1,4,7, after crossover operation, $Ya$ becomes 7,5,6,2,4,3,1,1,4,7 and $Yb$ becomes 2,5,4,3,7,1,6,2,5,7 (see Figure 5).

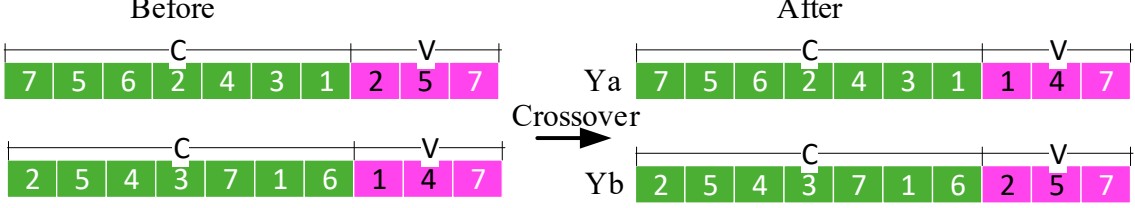

**Figure 5.** Crossover operation.

### 5.2.5. Mutation Operation

Define beforehand a parameter $p\_m$, which is regarded as the mutation probability and do the next process for $k = 1, 2, \ldots, popsize$: Generate a random number r in the unit interval [0,1], and select the chromosome $Yk$ to perform the mutation operation if $r < p\_c$.

The mutation operation acts on part C, take the chromosome $Yk$, for example, randomly generate two integers $a, b \in (0, length(Part\,C)]$ (see red arrows in Figure 6), then exchange the numbers the arrow points to. Before the mutation operation, the chromosome $Yk$ is coded as 7,5,6,2,4,3,1,2,5,7, then it becomes 7,5,3,2,4,6,1,2,5,7, the whole mutation operation is shown in Figure 6.

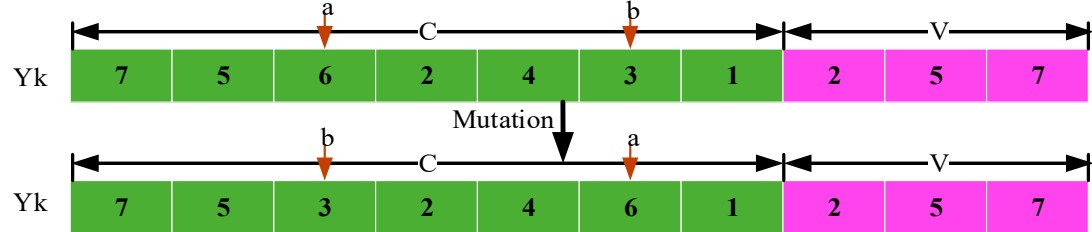

**Figure 6.** Mutation operation.

Figure 7 shows the whole process to solve the problem. The procedure of aforementioned fuzzy random simulation-based GA is as follows:

Step 1. Set $i = 1$. Initialize relative parameters, such as population size, maximum generation, crossover probability, mutation probability, and so on.

Step 2. Randomly generate *pop_size* feasible chromosomes to be the initial population.

Step 3. According to the models used, calculate the objective values for all chromosomes to obtain evaluation values by fuzzy random simulation.

Step 4. After performing selection, crossover, and mutation operations, update all chromosomes.

Step 5. If $i = max\_gen$, the simulation must be terminated, then choose the best chromosome as the ultimate solution. Otherwise, set $i = i + 1$ and return to Step 3.

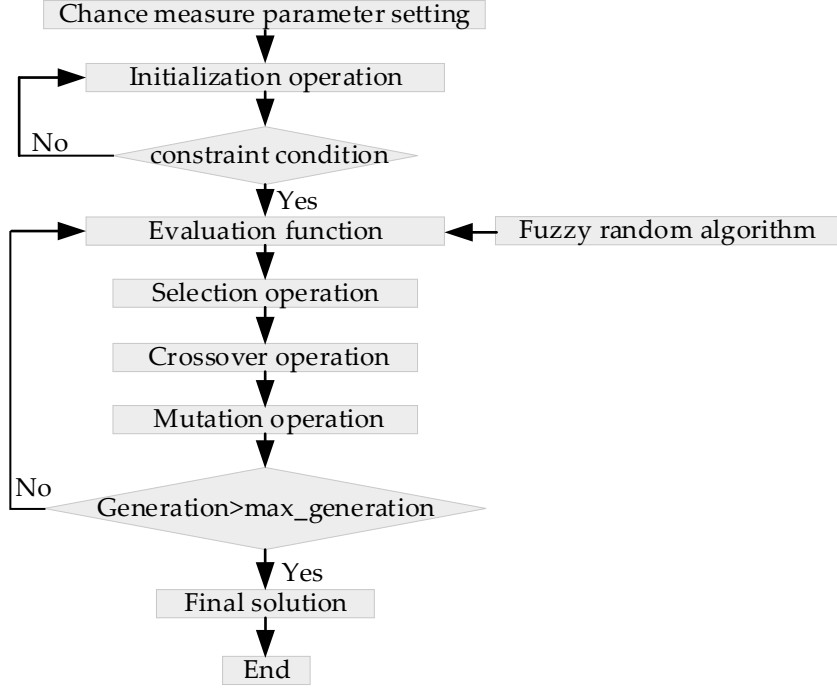

**Figure 7.** Hybrid intelligent algorithm flow chart.

As can be seen from Figure 7, the calculation process of the three models is the same. The difference lies in the evaluation function acquisition. In order to make the model proposed in this paper more applicable, the design of chromosomes needs to traverse all vehicle path combinations, according to the number of fleets and the load requirements of vehicles. After setting different chance level values, the objective values between each pair of nodes in the path need to be simulated for N times, which was set as 5000 in this study. Therefore, each operation of the genetic algorithm, be it crossover or mutation, needs to be repeated for 5000× N2 times, so the complexity of the algorithm is O(N2). Not only this, it still needs to satisfy the limits of capacity limitation, as shown in Section 4.4, so it is more complex than the model in the deterministic environment.

## 6. Case Study

In this section, two numerical experiments are presented to illustrate the efficiency of the proposed three models and solution methodology. The first one was a small-size network that consisted of only 10 customers, while the second case considered a practical case which was applied to the Changchun city, Jilin province in China. When calculating the objectives of Model II and III, for the specific values of the parameters of cost refer to Table 3.

**Table 3.** Explanation and value for parameters in Equation 9.

| Parameter | Meaning | Value |
|-----------|---------|-------|
| $M$ | Total quality | 5 t |
| $\theta$ | Road angle of arc | 0 |
| $a$ | Acceleration | 0 |
| $g$ | Gravitational constant | $9.8\,\mathrm{m/s}^2$ |
| $C_d$ | Drag resistance coefficient | 0.7 |
| $C_h$ | Rolling resistance coefficient | 0.01 |
| $\zeta$ | Air density | $1.2041\ \mathrm{kg/m}^2$ |
| $S$ | Surface area of vehicle | $5\ \mathrm{m}^2$ |
| $P_{fuel}$ | Fuel price | 7 RMB/L |
| $\eta_c$ | Fuel conversion factor | $2.32\,\mathrm{kg/L}$ |
| $t_c$ | carbon tax | $0.6\,\mathrm{RMB/kg}$ |
| $\vartheta$ | Fuel efficiency | 20% |

### 6.1. Case 1: Small Case

In this case, a small-scale network was established, and it contained one deport D0 and 10 customers named C1, . . . , C10 are shown as Figure 8. The demand amounts of customers were scheduled to be t = (2.5, 4.0, 4.6, 3.2, 2.8, 3.8, 3.0, 3.0, 3.0, 2.0) t. The capacity of the depot was assumed to be 40 t, the arc length and population density were described in the second and third column of Table 4. Then, the random accident probability and driving speed were shown in the third and fourth column of Table 4. The last column in Table 4 was the affected area, once the accident happened. The length of arc obeyed equipossible fuzzy distribution, while the population density was also a fuzzy triangle variable. Two random variables—accident probability and driving speed, the former was normally distributed, while the latter was uniformly distributed. Here, all fuzzy parameters and random variables were assumed to be independent.

**Table 4.** The risk and cost in Example 6.1.

| Arc | Length (km) | Population Density (pop/km$^2$) | Pr ($\times 10^{-5}$) | Speed (km/h) | Area (km$^2$) |
|---|---|---|---|---|---|
| (0,1) | (70,90) | (80,83,87) | N(1,9) | U(40,60) | 24 |
| (0,2) | (100,120) | (185,189,205) | N(1,4) | U(40,60) | 32 |
| (0,3) | (90,110) | (200,210,235) | N(2,9) | U(40,60) | 18 |
| (0,4) | (75,90) | (70,72,75) | N(2,16) | U(40,60) | 27 |
| (0,5) | (150,180) | (140,145,160) | N(3,9) | U(40,60) | 56 |
| (0,6) | (200,230) | (65,70,72) | N(1,1) | U(40,60) | 60 |
| (0,7) | (40,60) | (260,270,278) | N(1,4) | U(40,60) | 19 |
| (0,8) | (120,140) | (262,268,270) | N(4,9) | U(40,60) | 45 |
| (0,9) | (220,250) | (205,210,220) | N(1,9) | U(40,60) | 52 |
| (0,10) | (120,140) | (150,160,165) | N(1,6) | U(40,60) | 38 |
| (1,2) | (100,150) | (78,81,85) | N(1,5) | U(40,60) | 29 |
| (1,3) | (220,300) | (65,72,88) | N(2,9) | U(40,60) | 41 |
| (1,4) | (150,330) | (210,220,235) | N(3,16) | U(40,60) | 37 |
| (1,5) | (210,300) | (270,278,298) | N(1,2) | U(40,60) | 87 |
| (1,6) | (180,260) | (68,70,72) | N(2,2) | U(40,60) | 34 |
| (1,7) | (110,350) | (130,135,140) | N(2,4) | U(40,60) | 15.5 |
| (1,8) | (180,280) | (160,170,180) | N(3,9) | U(40,60) | 21.5 |
| (1,9) | (160,300) | (300,310,322) | N(3,1) | U(40,60) | 40 |
| (1,10) | (220,300) | (96,100,105) | N(3,2) | U(40,60) | 31 |
| (2,3) | (115,135) | (100,105,115) | N(3,4) | U(40,60) | 17 |
| (2,4) | (290,340) | (102,110,124) | N(1,16) | U(40,60) | 29 |
| (2,5) | (260,300) | (165,175,186) | N(2,16) | U(40,60) | 47 |
| (2,6) | (180,240) | (320,336,361) | N(2,9) | U(40,60) | 16 |
| (2,7) | (260,380) | (280,295,315) | N(3,5) | U(40,60) | 25 |
| (2,8) | (220,235) | (90,95,100) | N(4,9) | U(40,60) | 82 |
| (2,9) | (220,380) | (200,205,220) | N(4,1) | U(40,60) | 54 |
| (2,10) | (300,360) | (55,60,62) | N(4,2) | U(40,60) | 26 |
| (3,4) | (180,380) | (170,175,186) | N(4,3) | U(40,60) | 54 |
| (3,5) | (120,280) | (215,222,236) | N(4,4) | U(40,60) | 61 |
| (3,6) | (200,260) | (105,108,116) | N(5,9) | U(40,60) | 57 |
| (3,7) | (210,290) | (195,203,215) | N(5,1) | U(40,60) | 38 |
| (3,8) | (160,390) | (155,162,176) | N(5,2) | U(40,60) | 28 |
| (3,9) | (280,360) | (195,202,215) | N(5,4) | U(40,60) | 37 |
| (3,10) | (160,420) | (66,68,72) | N(5,6) | U(40,60) | 26 |
| (4,5) | (220,270) | (305,315,340) | N(5,16) | U(40,60) | 45 |
| (4,6) | (260,350) | (345,350,370) | N(6,9) | U(40,60) | 32 |
| (4,7) | (120,210) | (210,215,230) | N(6,1) | U(40,60) | 17.5 |
| (4,8) | (180,240) | (75,78,82) | N(6,2) | U(40,60) | 47.2 |
| (4,9) | (240,350) | (55,60,64) | N(6,3) | U(40,60) | 35 |
| (4,10) | (220,310) | (320,354,376) | N(6,4) | U(40,60) | 43 |
| (5,6) | (190,285) | (310,322,340) | N(6,1) | U(40,60) | 34 |
| (5,7) | (180,320) | (280,290,312) | N(6,4) | U(40,60) | 28 |
| (5,8) | (320,420) | (260,285,320) | N(6,2) | U(40,60) | 34 |
| (5,9) | (250,360) | (115,119,128) | N(1,1) | U(40,60) | 64 |
| (5,10) | (170,250) | (175,180,192) | N(1,2) | U(40,60) | 35 |
| (6,7) | (220,290) | (260,275,292) | N(1,3) | U(40,60) | 29 |
| (6,8) | (270,365) | (220,231,245) | N(1,4) | U(40,60) | 75 |
| (6,9) | (320,390) | (293,300,321) | N(2,9) | U(40,60) | 61 |
| (6,10) | (305,375) | (215,223,236) | N(2,1) | U(40,60) | 84 |
| (7,8) | (260,320) | (293,303,324) | N(2,3) | U(40,60) | 34 |
| (7,9) | (150,240) | (200,210,225) | N(2,4) | U(40,60) | 43 |
| (7,10) | (210,275) | (262,272,286) | N(2,5) | U(40,60) | 19 |
| (8,9) | (200,255) | (360,383,405) | N(3,9) | U(40,60) | 56 |
| (8,10) | (230,285) | (260,266,282) | N(3,4) | U(40,60) | 34 |
| (9,10) | (180,270) | (365,377,400) | N(3,1) | U(40,60) | 28 |

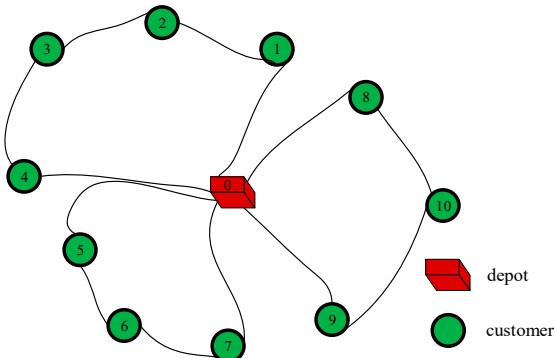

**Figure 8.** Small case for vehicle routing problem.

First, the authors compared the best solution with different chromosome population, it performed hybrid intelligence GA algorithm with *pop_size* = 3, 30, 100 and *max_generation* = 50, respectively. The crossover probability was set to be 0.8 while the mutation probability was defined as 0.3. To increase the credibility measure of the best solution, the chance level, that is to say, the credibility level (Cr) and the probability level (Pr) were both set to be 0.99. The result generated by the hybrid intelligence algorithm for Model I, Model II, and Model III are presented in Table 5, Table 6, Table 7, respectively. For example, as for Model I, when the *pop size* was 30, the optimum vehicle routing arrangement was:

Vehicle 1: 0-5-2-1-7-8-4-0

Vehicle 2: 0-9-0

Vehicle 3: 0-10-3-6-0

and the sum risk was 7.3103, simulation time was 8197.9 s. The larger the chromosome population, the smaller was the risk, and the more simulation time it consumed. It also showed that the proposed model and algorithm were feasible.

Second, Model II and Model III were simulated under the same conditions, it also conformed to the result that the lower the objective value, the greater the consumption time. It was obvious that this was an important problem that time complexity increased exponentially, which needs to be solved in the long run.

Last but not least, Model III integrates risk and cost, the weight for risk and cost is set to be 0.7 and 0.3, whatever is the weight, the models and algorithm could generate a reasonable solution.

**Table 5.** Model I result by hybrid intelligence algorithm for case 1.

| Genetic Algorithm Parameters | | | Chance Level | Model I | | |
|---|---|---|---|---|---|---|
| **pop** | **Max_Gen** | **pc pm** | **Cr Pr** | **Best Solution** | **Risk** | **Consume Time** |
| 3 | 50 | 0.8 0.3 | 0.99 0.99 | 0-1-6-0 <br> 0-2-9-5-4-8-10-0 <br> 0-7-3-0 | 8.7891 | 1203.68 s |
| 30 | 50 | 0.8 0.3 | 0.99 0.99 | 0-5-2-1-7-8-4-0 <br> 0-9-0 <br> 0-10-3-6-0 | 7.3103 | 8197.9 s |
| 100 | 50 | 0.8 0.3 | 0.99 0.99 | 0-5-3-10-8-4-0 <br> 0-7-9-2-6-1-0 | 6.6191 | 24,294.5 s |

**Table 6.** Model II result by hybrid intelligence algorithm for case 1.

| Genetic Algorithm Parameters | | pc | Chance Level Cr | Model II | | |
|---|---|---|---|---|---|---|
| pop | Max gen | pm | Pr | Best Solution | Cost | Consume Time |
| 3 | 50 | 0.8 0.3 | 0.99 0.99 | 0-8-9-0 <br> 0-10-7-4-3-1-6-0 <br> 0-5-2-0 | 38,393.4 | 768.761 s |
| 30 | 50 | 0.8 0.3 | 0.99 0.99 | 0-1-7-8-3-2-1 <br> 0-4-5-10-9-0 <br> 0-6-0 | 34,542.4 | 2882.98 s |
| 100 | 50 | 0.8 0.3 | 0.99 0.99 | 0-1-6-0 <br> 0-2-9-5-4-8-10-0 <br> 0-7-3-0 | 30,799.8 | 9184.34 s |

**Table 7.** Model III result by hybrid intelligence algorithm for case 1.

| Genetic Algorithm Parameters | | pc pm | Chance Level Cr Pr | Model III | | |
|---|---|---|---|---|---|---|
| pop | Max gen | | | Best Solution | 0.7 Risk + 0.3 Cost | Consume Time |
| 3 | 50 | 0.8 0.3 | 0.99 0.99 | 0-8-9-10-7-4-0 <br> 0-3-1-0 <br> 0-6-5-2-0 | 0.84532 | 768.761 s |
| 30 | 50 | 0.8 0.3 | 0.99 0.99 | 0-9-2-6-10-0 <br> 0-8-0 <br> 0-7-3-5-1-4-0 | 0.832983 | 4843.76 s |
| 100 | 50 | 0.8 0.3 | 0.99 0.99 | 0-1-6-0 <br> 0-2-9-5-4-8-10-0 <br> 0-7-3-0 | 0.792522 | 15824.5 s |

*6.2. Case 2: Practical Case*

In this example, a practical case—gas transportation in the Changchun City, Jilin province, China was considered. The topological graph is shown in Figure 9, and it consisted of 15 customers named from node 1 to node 15 and one depot called depot 0. It was located in the northern part of the city and the main road is shown as the black lines. The length, population density, probability, speed, and affected area are presented from column 2 to column 6 in Table 8. The difference from case 1 was that if the arc length was Inf, it implied that there was no connection between the two customers.

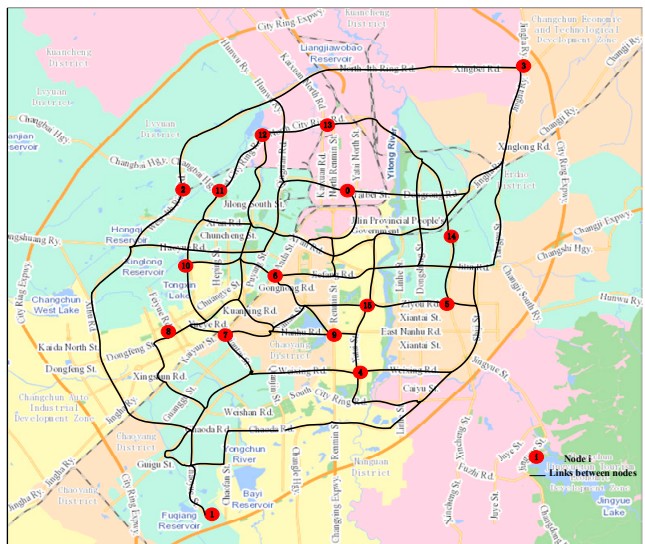

**Figure 9.** Practical case for vehicle routing problem.

**Table 8.** The risk and cost in Example 6.2.

| Arc | Length (km) | Population Density (pop per km²) | Probability (×10⁻⁵) | Speed (km/h) | Area (km²) |
|---|---|---|---|---|---|
| (0,1) | Inf | Inf | N(1,2) | U(40,60) | Inf |
| (0,2) | (10,15) | (15,20,38) | N((1,3) | U(40,60) | 2 |
| (0,3) | (15,20) | (20,30,40) | N(2,6) | U(40,60) | 5 |
| (0,4) | (30,40) | (20,33,44) | N(1,4) | U(40,60) | 6 |
| (0,5) | (15,30) | (22,35,50) | N(1,6) | U(40,60) | 3 |
| (0,6) | (10,12) | (18,20,32) | N(2,4) | U(40,60) | 1 |
| (0,7) | (8,15) | (26,28,29) | N(1,1) | U(40,60) | 3.5 |
| (0,8) | (18,24) | (20,25,30) | N(2,5) | U(40,60) | 4.8 |
| (0,9) | (10,13) | (15,20,28) | N(1,3) | U(40,60) | 1.6 |
| (0,10) | (10,18) | (15,20,24) | N(2,5) | U(40,60) | 1.7 |
| (0,11) | (8,12) | (20,26,27) | N(1,4) | U(40,60) | 2.1 |
| (0,12) | (5,10) | (10,16,30) | N(1,1) | U(40,60) | 2.3 |
| (0,13) | (6,14) | (30,32,35) | N(2,1) | U(40,60) | 2.6 |
| (0,14) | (7,16) | (22,28,40) | N(2,2) | U(40,60) | 3 |
| (0,15) | (8,15) | (12,20,26) | N(1,3) | U(40,60) | 2.9 |
| (1,2) | (40,42) | (50,60,70) | N(2,6) | U(40,60) | 14 |
| (1,3) | (60,62) | (60,70,85) | N(2,8) | U(40,60) | 22 |
| (1,4) | (20,25) | (32,38,40) | N(2,3) | U(40,60) | 7 |
| (1,5) | (20,30) | (28,29,36) | N(2,4) | U(40,60) | 7.5 |
| (1,6) | (25,28) | (40,42,50) | N(3,5) | U(40,60) | 8 |
| (1,7) | (15,18) | (20,24,30) | N(1,4) | U(40,60) | 3 |
| (1,8) | (8,12) | (20,25,30) | N(2,2) | U(40,60) | 2.6 |
| (1,9) | Inf | Inf | Inf | U(40,60) | Inf |
| (1,10) | (28,32) | (39,45,48) | N(2,5) | U(40,60) | 10 |
| (1,11) | (20,26) | (30,36,45) | N(2,6) | U(40,60) | 8.2 |
| (1,12) | Inf | Inf | Inf | U(40,60) | Inf |
| (1,13) | (80,90) | (100,102,110) | N(1,5) | U(40,60) | 20 |
| (1,14) | (70,80) | (80,82,88) | N(3,6) | U(40,60) | 15 |
| (1,15) | (30,35) | (40,42,48) | N(1,3) | U(40,60) | 12.5 |
| (2,3) | (20,25) | (30,38,40) | N(1,1) | U(40,60) | 10.2 |
| (2,4) | (40,50) | (42,45,48) | N(3,3) | U(40,60) | 14.5 |
| (2,5) | (60,70) | (66,68,70) | N(3,7) | U(40,60) | 18 |
| (2,6) | Inf | Inf | Inf | U(40,60) | Inf |
| (2,7) | Inf | Inf | Inf | U(40,60) | Inf |
| (2,8) | (50,55) | (65,70,78) | N(5,2) | U(40,60) | 22 |

**Table 8.** *Cont.*

| Arc | Length (km) | Population Density (pop per km²) | Probability (×10⁻⁵) | Speed (km/h) | Area (km²) |
|---|---|---|---|---|---|
| (2,9) | Inf | Inf | Inf | U(40,60) | Inf |
| (2,10) | (5,8) | (10,15,20) | N(1,2) | U(40,60) | 1.2 |
| (2,11) | (10,15) | (12,15,19) | N(2,2) | U(40,60) | 1.8 |
| (2,12) | (20,25) | (30,32,37) | N(2,3) | U(40,60) | 2.0 |
| (2,13) | (25,30) | (40,46,50) | N(2,4) | U(40,60) | 2.3 |
| (2,14) | Inf | Inf | Inf | U(40,60) | Inf |
| (2,15) | Inf | Inf | Inf | U(40,60) | Inf |
| (3,4) | (40,50) | (50,55,60) | N(3,2) | U(40,60) | 15 |
| (3,5) | (30,40) | (33,38,42) | N(3,5) | U(40,60) | 14.6 |
| (3,6) | (30,35) | (32,39,49) | N(2,5) | U(40,60) | 13.2 |
| (3,7) | (70,75) | (82,84,90) | N(5,8) | U(40,60) | 30 |
| (3,8) | (80,82) | (90,95,105) | N(4,9) | U(40,60) | 32 |
| (3,9) | Inf | Inf | Inf | U(40,60) | Inf |
| (3,10) | (60,70) | (65,70,80) | N(3,5) | U(40,60) | 23 |
| (3,11) | (40,50) | (50,62,82) | N(3,3) | U(40,60) | 24 |
| (3,12) | (40,42) | (50,62,72) | N(3,6) | U(40,60) | 26 |
| (3,13) | (35,40) | (42,50,53) | N(3,5) | U(40,60) | 25 |
| (3,14) | (15,22) | (32,38,42) | N(2,3) | U(40,60) | 10.2 |
| (3,15) | (25,30) | (30,40,42) | N(3,2) | U(40,60) | 20 |
| (4,5) | (15,20) | (20,25,36) | N(1,4) | U(40,60) | 16 |
| (4,6) | (30,40) | (42,55,65) | N(2,2) | U(40,60) | 25 |
| (4,7) | (10,15) | (20,26,34) | N(1,3) | U(40,60) | 6.5 |
| (4,8) | (15,25) | (20,28,35) | N(1,2) | U(40,60) | 7.2 |
| (4,9) | (5,8) | (10,20,32) | N(1,1) | U(40,60) | 2.5 |
| (4,10) | (50,80) | (65,72,88) | N(3,4) | U(40,60) | 19 |
| (4,11) | (60,62) | (80,82,86) | N(4,5) | U(40,60) | 28 |
| (4,12) | (50,55) | (62,68,72) | N(2,4) | U(40,60) | 24 |
| (4,13) | (40,50) | (44,48,52) | N(1,6) | U(40,60) | 20 |
| (4,14) | (15,20) | (20,25,30) | N(2,2) | U(40,60) | 13 |
| (4,15) | (4,15) | (10,20,32) | N(1,1) | U(40,60) | 2.2 |
| (5,6) | (8,12) | (15,20,25) | N(1,2) | U(40,60) | 4.5 |
| (5,7) | (12,15) | (20,23,27) | N(1,3) | U(40,60) | 7.2 |
| (5,8) | (13,20) | (20,30,40) | N(1,3) | U(40,60) | 7.5 |
| (5,9) | Inf | Inf | Inf | U(40,60) | Inf |
| (5,10) | (60,70) | (80,90,96) | N(2,5) | U(40,60) | 26 |
| (5,11) | (70,72) | (82,88,90) | N(2,6) | U(40,60) | 32 |
| (5,12) | Inf | Inf | Inf | U(40,60) | Inf |
| (5,13) | (30,50) | (36,44,54) | N(2,3) | U(40,60) | 15 |
| (5,14) | (8,10) | (20,22,28) | N(1,3) | U(40,60) | 4.2 |
| (5,15) | (5,12) | (12,18,28) | N(1,2) | U(40,60) | 2.4 |
| (6,7) | (5,8) | (13,18,28) | N(1,1) | U(40,60) | 2.2 |
| (6,8) | (8,12) | (20,32,40) | N(1,2) | U(40,60) | 4 |
| (6,9) | (8,12) | (20,32,40) | N(1,2) | U(40,60) | 4 |
| (6,10) | (8,12) | (20,32,40) | N(1,2) | U(40,60) | 4 |
| (6,11) | (10,15) | (25,30,38) | N(1,3) | U(40,60) | 4.2 |
| (6,12) | (12,15) | (25,32,42) | N(1,4) | U(40,60) | 4.4 |
| (6,13) | (15,18) | (20,30,40) | N(2,2) | U(40,60) | 4.8 |
| (6,14) | (16,20) | (22,30,32) | N(2,2) | U(40,60) | 5 |
| (6,15) | (8,10) | (15,20,22) | N(1,3) | U(40,60) | 3 |
| (7,8) | (5,7) | (12,20,22) | N(1,1) | U(40,60) | 1.9 |
| (7,9) | (8,12) | (18,24,35) | N(1,2) | U(40,60) | 2.8 |
| (7,10) | (10,14) | (25,30,33) | N(1,3) | U(40,60) | 4.3 |
| (7,11) | (15,20) | (22,28,32) | N(2,1) | U(40,60) | 4.5 |
| (7,12) | (16,22) | (23,29,35) | N(2,2) | U(40,60) | 4.6 |
| (7,13) | (20,22) | (30,36,40) | N(2,3) | U(40,60) | 4.8 |
| (7,14) | (20,30) | (30,36,45) | N(2,3) | U(40,60) | 5.6 |
| (7,15) | (10,12) | (17,22,25) | N(1,1) | U(40,60) | 3.8 |

**Table 8.** *Cont.*

| Arc | Length (km) | Population Density (pop per km²) | Probability (×10⁻⁵) | Speed (km/h) | Area (km²) |
|---|---|---|---|---|---|
| (8,9) | (10,12) | (17,22,25) | N(1,1) | U(40,60) | 3.8 |
| (8,10) | (9,12) | (10,15,16) | N(1,2) | U(40,60) | 3.6 |
| (8,11) | (15,25) | (12,16,23) | N(2,1) | U(40,60) | 5.2 |
| (8,12) | (15,25) | (12,16,23) | N(2,1) | U(40,60) | 5.2 |
| (8,13) | (15,30) | (12,18,28) | N(2,3) | U(40,60) | 5.6 |
| (8,14) | (35,40) | (50,55,62) | N(2,3) | U(40,60) | 28 |
| (8,15) | (10,12) | (20,22,25) | N(1,1) | U(40,60) | 4.8 |
| (9,10) | (8,12) | (15,20,25) | N(1,1) | U(40,60) | 2.2 |
| (9,11) | (12,15) | (20,28,30) | N(1,2) | U(40,60) | 3.3 |
| (9,12) | (15,18) | (25,30,32) | N(1,2) | U(40,60) | 3.3 |
| (9,13) | (15,20) | (25,30,36) | N(1,2) | U(40,60) | 3.0 |
| (9,14) | (10,12) | (20,22,25) | N(1,1) | U(40,60) | 2.6 |
| (9,15) | (4,8) | (10,20,30) | N(0,1) | U(40,60) | 1 |
| (10,11) | (4,6) | (10,20,26) | N(0,1) | U(40,60) | 0.9 |
| (10,12) | (8,10) | (10,26,32) | N(1,3) | U(40,60) | 2.0 |
| (10,13) | (10,15) | (18,28,36) | N(1,2) | U(40,60) | 3.2 |
| (10,14) | (18,20) | (20,30,40) | N(2,1) | U(40,60) | 3.1 |
| (10,15) | (10,12) | (18,28,32) | N(2,1) | U(40,60) | 2.6 |
| (11,12) | (5,8) | (15,20,25) | N(1,1) | U(40,60) | 0.8 |
| (11,13) | (8,10) | (18,22,28) | N(1,2) | U(40,60) | 1.1 |
| (11,14) | (15,18) | (20,32,38) | N(1,3) | U(40,60) | 3.3 |
| (11,15) | (12,14) | (20,24,28) | N(2,3) | U(40,60) | 2.8 |
| (12,13) | (6,8) | (10,16,22) | N(1,1) | U(40,60) | 0.6 |
| (12,14) | (10,15) | (20,28,32)) | N(1,2) | U(40,60) | 2.7 |
| (12,15) | (12,18) | (18,26,31) | N(2,1) | U(40,60) | 2.8 |
| (13,14) | (12,14) | (16,20,26) | N(2,1) | U(40,60) | 2.8 |
| (13,15) | (20,25) | (30,32,38) | N(3,1) | U(40,60) | 5.2 |
| (14,15) | (10,12) | (18,22,26) | N(2,2) | U(40,60) | 2.0 |

Similarity with case I, using Algorithm 5.1—Hybrid intelligence algorithm, authors obtained the vehicle routing solution in column 7 from Tables 9–11, for the three models.

**Table 9.** Model I results by hybrid intelligence algorithm for Case 2.

| Genetic Algorithm Parameters | | | Chance Level | Model I | | |
|---|---|---|---|---|---|---|
| pop | Max_gen | pc pm | Cr Pr | Best Solution | Risk | Consume Time |
| 3 | 50 | 0.8 0.3 | 0.99 0.99 | 0-12-13-15-0<br>0-10-6-4-5-1-2-11-14-9-8-0<br>0-7-3-0 | 0.571448 | 296.519 s |
| 30 | 50 | 0.8 0.3 | 0.99 0.99 | 0-4-14-5-13-0<br>0-10-1-2-6-0<br>0-7-9-8-11-15-12-2-0 | 0.372853 | 2051.57 s |
| 100 | 50 | 0.8 0.3 | 0.99 0.99 | 0-8-6-11-0<br>0-10-1-2-13-12-9-15-3-0<br>0-14-7-0 | 0.129406 | 7010.85 s |

**Table 10.** Model II results by hybrid intelligence algorithm for Case 2.

| Genetic Algorithm Parameters | | | Chance Level | Model II | | |
|---|---|---|---|---|---|---|
| pop | Max_gen | pc<br>pm | Cr<br>Pr | Best Solution | Cost | Consume Time |
| 3 | 50 | 0.8<br>0.3 | 0.99<br>0.99 | 0-12-13-15-0<br>0-10-6-4-5-1-2-11-14-9-8-0<br>0-7-3-0 | 3658.17 | 579.293 s |
| 30 | 50 | 0.8<br>0.3 | 0.99<br>0.99 | 0-8-6-0<br>0-11-10-1-2-0<br>0-13-12-9-15-3-5-4-14-7-0 | 834.382 | 3972.98 s |
| 100 | 50 | 0.8<br>0.3 | 0.99<br>0.99 | 0-8-6-0<br>0-11-10-1-2-0<br>0-13-12-9-15-3-5-4-14-7-0 | 820.419 | 15,353.4 s |

**Table 11.** Model III results by hybrid intelligence algorithm for Case 2.

| Genetic Algorithm Parameters | | | Chance Level | Model III | | |
|---|---|---|---|---|---|---|
| pop | Max_gen | pc<br>pm | Cr<br>Pr | Best Solution | 0.7Risk +<br>0.3 Cost | Consume Time |
| 3 | 50 | 0.8<br>0.3 | 0.99<br>0.99 | 0-1-9-3-0<br>0-13-8-6-14-12-11-5-2-15-7-0<br>0-10-4-0 | 0.982762 | 901.49 s |
| 30 | 50 | 0.8<br>0.3 | 0.99<br>0.99 | 0-12-15-3-14-13-0<br>0-10-0<br>0-4-11-8-2-5-9-1-6-7-0 | 0.803568 | 6395.4 s |
| 100 | 50 | 0.8<br>0.3 | 0.99<br>0.99 | 0-9-14-0<br>0-12-5-6-8-1-10-0<br>0-4-3-11-7-15-2-13-0 | 0.778511 | 20,173 s |

As shown by the simulation results, the authors could draw a conclusion that although the size of the case increased, the feasible solution was still obtained. For the sake of demonstrating the efficiency for the proposed algorithm, it selected *pop_size* = 100, *max_gen* = 50 for the three models. To compare and analyze the algorithm convergence, the results are shown in Figure 10.

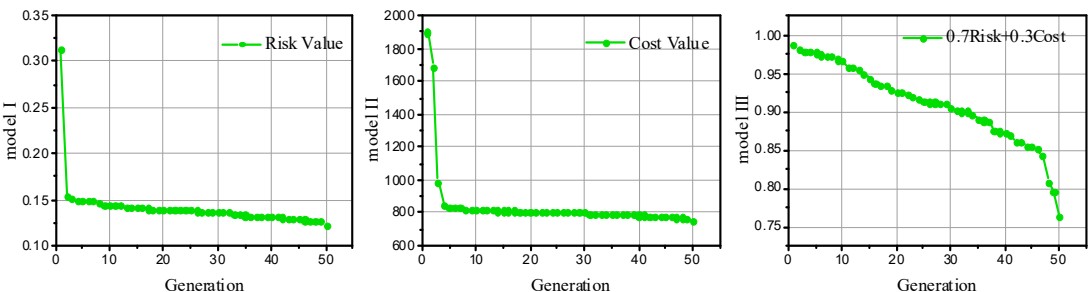

**Figure 10.** Convergence of GA for the three models.

Note that the convergence curve trend of model I and model II was same, in the first few generations, the decline was rapid, then optimization process was extremely slow from the 5th to 50th generation. This was because, as the number of simulations in each generation was up to 3000, it could get the optimal value. However, for model III, it could be optimized slowly because it calculated

the normalized values for risk and cost. The curves demonstrated the algorithm feasibility from the computing angle.

The two cases are coded in C++ language, using software Visual Studio 2012, performed on a personal computer with an Intel(R) core i5 and 12G RAM, the total simulation time of case 1 and case 2 were 19 h and 16 h, respectively. In line with this, with an increase in the chromosome population and customer scale, the simulation time showed a clear exponential growth. Note that no matter what the chance level defined, it could present a reasonable solution in the end. In spite of the proposed algorithm being time-consuming, since it spent most time on the chromosome initialization, it could remarkably bring down objective values, and provide best solutions to participants in the supply chain.

## 7. Conclusions

Based on the typical characteristics (random and fuzzy) of risk and cost encountered during transportation, three chance-constrained programming models are presented. Additionally, in order to get more accurate simulation results, a hybrid intelligence algorithm integrating the fuzzy random algorithm and GA algorithm was designed. Finally, the model performance was verified by two numerical cases. The major contributions of this study are summarized as follows:

The risk model combining probability measure and credibility measure was developed for hazmat transportation. Most risk assessment models applied to hazmat so far see risk occurring as a stochastic event, which might lead to the prevention of accident, regardless of accident consequence, and might cause a terrible scenario happen. This study considered the ace length and population density as a fuzzy variable, and thus, modified the traditional risk from a practical point of view.

(1) The VRP models using uncertain theory were established by taking risk, cost, risk and cost, as the objective function for hazmat transportation, respectively. According to the risk assessment model, VRP models must be extended to chance-constrained. Different chance level has different solutions for the decision-maker, this can respond to changes in vehicle routing arrangements in time, when an accident occurs.

(2) In order to attain feasible solution for the models proposed by this paper, a hybrid intelligence algorithm integrating the fuzzy random algorithm and GA algorithm was designed. It included mass of simulation calculation; however, two numerical cases showed that the models were efficient, and the hybrid intelligent algorithm was steady, convergent for a small-size, and a middle-size problem.

Despite getting a reasonable optimal solution, it took too long a time, for example, the longest time was up to 30,799.8 s, almost 8.5 h. It is necessary to seek more intelligent and time-saving algorithms to solve the problem. It also takes more time to use this model to solve the vehicle routing problem in large-scale scenarios, which is beyond the acceptance period of the participants. Additionally, when the hazmat transportation is applied to multiple depots scenario, the model in this study would be more complex, new intelligent algorithms might need developing, which are all possible future directions.

**Author Contributions:** Conceptualization, L.Z. and N.C.; Writing—original draft, methodology, investigation, L.Z.; Writing—review & editing, Funding acquisition, N.C. All authors have read and agreed to the published version of the manuscript.

**Funding:** This research was funded by Fundamental Research Funds for the Central Universities, under grant number 300102239103 and Young Teachers Research Project for the Xi'an University of Technology, under grant number 256081921.

**Conflicts of Interest:** The authors declare no conflict of interest.

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
