# Peer review of "Fuzzy Random Chance-Constrained Programming Model for the Vehicle Routing Problem of Hazardous Materials Transportation"

_symmetry, doi:10.3390/sym12081208_

Round 1

Reviewer 1 Report

I recommend the revision of the English language and the way of writing the paper.

The article contains only 4 keywords and this is not according to the Author’s Instructions.

I recommend passing of the paragraph 4.1. on the next page  (9).

I recommend passing the text presenting equations 18-22 ((paragraph 4.4, page 11)) before them.

I recommend passing the text (paragraph 6.2 Case 2: practical case) before figure 9.

The conclusions are short and do not present a comparative analysis of the characteristics and advantages of the three proposed models and those of the existing ones.

The bibliography contains 58 references but three of them ([13], [46] and [48]) are not current.

Reviewer 2 Report

The study is interesting and worth exploring. The authors articulate their work properly. My minor comments are:

Pleas discuss the research questions more elaborately in the introduction.

Replace 2.3 Summary with Research Gap. Also Highlight the contribution of this work there. The remainder of this paper is organized ….. This paragraph should be the last paragraph of Introduction section.

I will suggest the authors to include a research framework figure to understand the overall process effectively.

Please double check all the notations and define properly. For example: RMB/L, RMB/kg.

The conclusion section should be more precise in the case of presentation of scientific contributions and limitations to the work done.
